# Predicting Trauma Severity from Imbalanced Data Using Ensemble Regression and Generative Models

Saeka Rahman[1], Md Motiur Rahman[1], Elika Ridelman[2], Ashley Frei[2], Rebecca M. Adams[2], Mo Rastgaar[1],
Christina Shanti[2], and Miad Faezipour[1]*

[1]School of Engineering Technology, Purdue University, West Lafayette, IN, USA

[2]Department of Surgery, Division of Pediatric Surgery, Children's Hospital of Michigan/Wayne State University, Detroit, MI, USA

mfaezipo@purdue.edu*

*Abstract*—**The Injury Severity Score (ISS) is a critical metric in trauma care, widely used for assessing injury severity, guiding clinical decisions, and evaluating patient outcomes. Despite the practical challenges of computing the score, due to its clinical significance, we propose a machine learning (ML) framework to predict ISS using structured clinical, demographic, and vehicle data, routinely documented in trauma registries and hospitals. We evaluate four ensemble-based regression models such as Random Forest Regressor (RFR), Gradient Boosting Regressor (GBR), eXtreme Gradient Boosting (XGBoost) regressor, and Gradient Boosting Machine (LightGBM). We identify GBR as the best performer when applied on a dataset generated at our clinical site, with coefficient of determination $R^2 = 0.78$. However, the original dataset exhibits substantial imbalance, with most cases concentrated in low-severity scores. To address the challenges of skewed ISS distribution, we implement various data augmentation techniques including transformations of target, resampling, interpolation, and noise-based strategies. Moreover, we develop two generative models Conditional Variational Autoencoder (cVAE) and Conditional Generative Adversarial Network (cGAN) to synthesize data from underrepresented severity ranges. The cVAE-augmented model achieves the highest performance of $R^2 = 0.94$, demonstrating the value of generative augmentation in enhancing regression accuracy under data imbalance.**

*Index Terms*—**Injury Severity Score (ISS), trauma severity, ensemble regression models, imbalanced data, data augmentation, generative models.**

## I. INTRODUCTION

Trauma is a significant global public health issue that is characterized by physical injury and subsequent psychological effects. This can have long-term implications on individuals' health and well-being [1]. Accurately assessing trauma severity is essential for clinical decision-making. Various scoring systems have been developed for this purpose. Among these, the Injury Severity Score (ISS) is particularly prominent due to its widespread clinical applicability and strong correlation with patient outcomes [2], [3]. It provides a standardized and reliable metric for evaluating trauma-related mortality and morbidity [4].

The ISS has the ability to inform early clinical decisions. Specifically, an ISS greater than 15 is indicative of significant trauma, requiring immediate and often advanced medical intervention [4]. Early assessment of trauma using ISS supports timely triage, treatment planning, and resource allocation in emergency settings [5]. Moreover, studies have shown that higher ISS scores are significantly associated with prolonged hospital stays and increased morbidity and mortality, emphasizing the importance of early and accurate prediction of trauma severity [6].

Despite the clinical significance, calculation of ISS typically requires detailed anatomical evaluation using various imaging techniques such as X-ray, computed tomography (CT), and ultrasound [7]. As a result, emergency physicians and trauma surgeons often have to make critical decisions based on limited initial observations without access to the full ISS. This limitation highlights the broader clinical need for predictive models capable of estimating ISS based on structured clinical and demographic data. One promising way for addressing this challenge is the application of machine learning (ML) techniques, which have demonstrated substantial potential across a wide range of trauma-related tasks.

Recent studies [8]–[12] have demonstrated the effectiveness of ML models in predicting in-hospital mortality, real-time deterioration, and 1-year post-trauma mortality. Beyond outcome prediction, ML tools have been applied to operational challenges, including transfusion prediction [13] and red blood cell demand estimation [14]. The application of ML extends to domain-specific contexts. In traumatic brain injury, artificial neural networks (ANNs) have outperformed traditional scoring systems such as Trauma and Injury Severity Score (TRISS) in mortality prediction tasks [12], [15].

Despite these advancements, direct ML-based prediction of ISS is underexplored. A recent study conducted by [16] developed neural machine translation (NMT) and feed-forward neural network (FFNN) models to estimate ISS from International Classification of Diseases (ICD) codes. Considering the importance of ISS in trauma assessment [17], the practical challenges of computing the score, and the potential of ML-driven solutions in this area, our research aims to develop a robust, data-driven model to predict ISS using structured clinical, demographic, and vehicle data that are collected from trauma registries or hospital records.

The main contributions of this research are as follows:

- We develop an ML framework for predicting ISS using a motor vehicle collision (MVC) dataset. The ML pipeline includes data pre-processing, feature engineering, model training, and performance evaluation. We implement and compare four ensemble-based regression models: Random Forest Regressor (RFR), Gradient Boosting Re-

gressor (GBR), eXtreme Gradient Boosting (XGBoost), and Light Gradient Boosting Machine (LightGBM), with GBR achieving the best performance ($R^2 = 0.78$).

- We demonstrate the skewness of the original dataset, highlighting the impact on the performance of the model. To mitigate the imbalance, we employ multiple augmentation methods, including transformation (logarithmic, sample weighting), resampling (Synthetic Minority Over-sampling Technique for Regression with Gaussian Noise (SMOGN), manual target-based upsampling, Kernel Density Estimation (KDE)-based upsampling), interpolation (mixup), and noise-based (Gaussian noise injection) techniques.

- We develop two deep learning-based generative models: Conditional Variational Autoencoder (cVAE) and Conditional GAN (cGAN) to synthesize high-quality samples from rare regions of the target space. The cVAE-augmented model achieves the highest predictive performance ($R^2 = 0.94$).

## II. METHODS

The overall workflow of our ISS prediction framework from raw data collection to final evaluation and augmentation is illustrated in Fig. 1. The process begins with the collection of raw crash data, followed by data pre-processing and feature engineering steps to prepare the dataset for the ML models. RFR, GBR, XGBoost regressor, and lightGBM regressor models are trained and evaluated. The performances are compared using Mean Absolute Error (MAE), Mean Squared Error (MSE), and coefficient of determination $R^2$. The best-performing model is identified and used for final predictions. To address the issue of data imbalance, various data augmentation strategies including transformation, resampling, interpolation, and noise-based augmentation, as well as generative models are used. The augmented dataset is fed into the best performing model, enhancing model robustness and improving generalization across the ISS spectrum.

### A. Dataset Description and Pre-processing

We have collected the motor vehicle collision (MVC) dataset at our level 1 verified trauma center at DMC (Detroit Medical Center), MI, USA. The original dataset consists of 1273 rows and 62 columns. Each instance comprises demographic attributes (e.g., age, gender, race), physiological measurements (e.g., blood pressure, pulse), injury attributes (e.g., type of injury, location of injury), treatment attributes (e.g., number of times the patient went to the operating room, hospital length of stay), collision attributes (e.g., location, type of vehicle, speed of vehicle) and the clinical severity score ISS. The ISS is computed by taking the square of the Abbreviated Injury Scale (AIS) scores from the top three injured regions, and summing them. Each AIS score ranges from 1 (minor) to 6 (unsurvivable). This provides a single composite score ranging from 1 to 75, where higher values indicate more severe trauma [18]. The formula to compute ISS is presented in Equation 1.

$$ISS = AIS_1^2 + AIS_2^2 + AIS_3^2 \qquad (1)$$

To ensure data quality and consistency, we apply several pre-processing steps. Columns with over 50% missing values are dropped, and missing numerical values are imputed with the mean, while categorical attributes are filled with the mode. We inspect the data for outliers, invalid entries, and column-wise inconsistencies to ensure all inputs are clean, numeric, and free of missing or infinite values.

### B. Feature Engineering

After cleaning and pre-processing the dataset we create three additional features named 'no injury', 'at least one injury', and 'more than one injury' in the dataset. Then, we perform feature importance analysis. Fig. 2 illustrates the feature importance graph showing the top 10 features ranked by importance along with other analyzed lower rank features, derived from the RFR trained to predict ISS. The results indicate that hospital length of stay (days) and intubated status are the most critical predictors, followed by age, vehicle speed, and certain discharge dispositions (e.g., discharged to morgue). Injury-specific features such as chest injury and abdominal injury also showed notable contributions. In contrast, demographic attributes including race, ethnicity, and less frequent discharge categories exhibited minimal impact on model predictions. We utilize 17 binary clinical features, 5 numerical clinical and demographic features, and 6 demographic and injury-related categorical features encoded using one-hot encoding (transforming each category into separate binary columns).

### C. Model Architectures

We employ ensemble model architectures such as RFR, GBR, XGBoost regressor, and LightGBM regressor to predict ISS. Though all are based on decision trees, they differ fundamentally in how trees are built and combined during training. The RFR is an ensemble of decision trees constructed using the bagging technique. Each tree is trained on a random subset of the data, and the final prediction is the average prediction of all trees [19]. The GBR builds trees sequentially, where each new tree corrects the residuals (errors) of the previous ensemble [20]. The model minimizes the loss function by fitting a new tree to the negative gradient of the loss. XGBoost is an advanced form of gradient boosting that adds regularization to improve generalization [21]. LightGBM optimizes gradient boosting by using histogram-based binning and a leaf-wise (instead of level-wise) tree growth strategy [22].

### D. Training Procedure

Our goal for this research is to learn a predictive function that maps structured clinical, demographic, and vehicle related features to a continuous target ISS. The dataset is split into training and testing subsets using stratified sampling to ensure a proportional representation of different severity levels across bins. Although the ISS is a continuous variable, it is discretized into five ordinal categories (0–10, 10–20, 20-30, 30-40, and greater than 40) to enable stratified sampling. This binning ensures that the training (80%) and testing (20%) sets retain similar patient severity distributions with 1017 samples in the training dataset and 255 in the testing dataset.

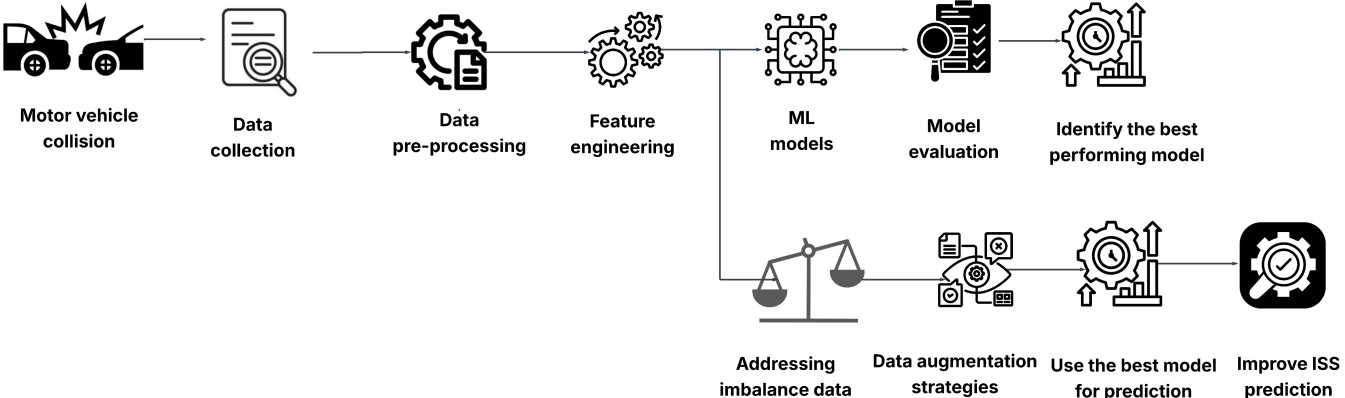

Fig. 1. Workflow diagram illustrating the ISS prediction pipeline using MVC data. The process starts with raw crash data collection, followed by preprocessing and feature engineering to prepare the dataset for ML models. The best-performing model is selected for final predictions. To address data imbalance, various augmentation strategies are applied, and the augmented dataset is fed into the selected model.

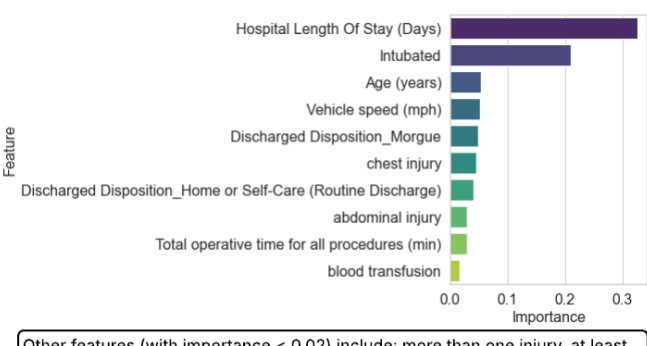

Other features (with importance < 0.02) include: more than one injury, at least one injury, pelvic injury, gender, head involvement, no injury, lower extremity injury, discharged to home, upper extrimity injury, face injury, burn, no. of procedures, spine injury, neck injury, surgery, race, ethnicity, skin graft

Fig. 2. Top 10 features ranked by importance along with a list of other lower rank features in the RFR model for ISS prediction. Importance is based on mean decrease in impurity across trees.

### E. Evaluation Metrics

We evaluate the performance of the models using three evaluation metrics; the Mean Absolute Error (MAE), Mean Squared Error (MSE), and the coefficient of determination $R^2$ [23]; presented in Equations 2, 3, and 4, correspondingly.

$$\text{MAE} = \frac{1}{N} \sum_{i=1}^{N} |y_i - \hat{y}_i| \qquad (2)$$

$$\text{MSE} = \frac{1}{N} \sum_{i=1}^{N} (y_i - \hat{y}_i)^2 \qquad (3)$$

$$R^2 = 1 - \frac{\sum_{i=1}^{N}(y_i - \hat{y}_i)^2}{\sum i = 1^N (y_i - \bar{y})^2} \qquad (4)$$

where $y_i$ is the actual value, $\hat{y}_i$ is the predicted value of the target variable for the $i^{\text{th}}$ observation, and $\bar{y}$ is the mean of the true target values.

### F. Imbalanced Data Handling

Medical datasets often suffer from right-skewed distributions in outcome variables such as severity or cost. This skewness causes regression models to underperform on rare but clinically significant high-severity cases. Data augmentation has emerged as a critical technique in the domain of

ML to enhance the performance of models trained on limited datasets. This section focuses on various methods to deal with the imbalanced data, including transformations, resampling, interpolation-based, noise-based, and generative models to augment the data.

*1) Transformations:* Logarithmic transformation aids in normalizing data distributions and mitigating the impact of outliers. It works by compressing skewed data, resulting in a distribution closer to normal and improving the statistical properties of the data [24]. We apply logarithmic transformation to the target variable (ISS) to compress the range of values and reduce skewness. In addition, we introduce sample weighting during model training [25], where greater weights are assigned to rare samples with higher ISS values. This encourages the model to place more learning emphasis on underrepresented and clinically critical cases.

*2) Resampling:*

*a) Synthetic Minority Over-sampling Technique for Regression with Gaussian Noise (SMOGN):* One prominent approach to augmenting tabular data is the SMOGN technique [26]. This method aims to boost datasets with imbalanced class distributions by synthetically generating new data points for minority classes based on the density of the available samples. Since regression does not have distinct class boundaries, SMOGN first defines a relevance function that quantifies the importance of each target value $y$ on a continuous scale from 0 to 1. SMOGN then applies under-sampling to reduce the dominance of the majority regions and over-sampling of rare regions using two techniques: (1) interpolation between minority samples, and (2) injection of Gaussian noise to promote diversity and reduce overfitting to synthetic samples. We utilize $k = 5$ nearest neighbors for synthetic instance generation.

*b) Target-based Upsampling:* We utilize two target-based upsampling techniques: manual and Kernel Density Estimation (KDE)-based. Manual target-based upsampling increases the density of rare target values by either duplicating existing observations or generating synthetic samples through interpolation [27]. In this approach, we define samples with

an ISS greater than 30 as rare and clinically critical. These high-severity cases are isolated to form a subset of the data. To amplify their presence in the training distribution, we randomly resample from this high-severity subset with replacement, generating three additional copies for each rare instance. This tripling of rare cases is achieved by concatenating the original dataset with a replicated sample drawn from the high-severity group. In KDE-based upsampling, instead of direct duplication, a probability density function is estimated over the rare samples using KDE, and new samples are drawn from this estimated distribution [27].

*3) Interpolation-based Augmentation:*

*a) Mixup Interpolation:* Mixup interpolation is inspired by data augmentation in deep learning. It involves creating new samples by linearly interpolating the input features and target values from random pairs of training instances [28]. In this approach, synthetic samples are generated by interpolating between pairs of randomly selected training instances. A mixing coefficient $\lambda$ is drawn from a Beta distribution (with shape parameter $\alpha = 0.3$) to control the interpolation strength. Each new feature vector is created as a weighted average of two real feature vectors, and the same interpolation is applied to their corresponding ISS values.

*4) Noise-based:*

*a) Gaussian Noise Injection:* Gaussian noise injection is another data augmentation approach where a small amount of noise is added to the input features of rare samples to synthetically expand the neighborhood around them [29]. This method maintains the original label while slightly altering the input space. To introduce controlled variability and expand the representation of rare high-severity cases (for ISS greater than 30), we apply Gaussian noise injection to their numeric features. Specifically, for each numeric feature, we calculate its standard deviation within the rare class and add noise sampled from a normal distribution with mean zero and standard deviation set to 5% of that feature's original standard deviation.

*5) Generative Models:*

*a) Conditional Variational Autoencoder (cVAE):* We implement cVAE model [30], a generative neural network architecture, designed for structured data augmentation, conditioned on ISS categories. The overall block diagram of the model is presented in Fig. 3. The model architecture consists of an encoder, a latent space with reparameterization, and a decoder. The input of the model includes a clinical feature vector and a one-hot encoded representation of the target ISS bin. These are concatenated and passed through the encoder. The encoder contains two fully connected layers with 64 and 32 units respectively, each followed by a Rectified Linear Unit (ReLU) activation.

From the encoder's output, two parallel linear layers produce the mean ($\mu$) and log-variance $\log \sigma^2$ of the latent Gaussian distribution. The latent variable is sampled using the reparameterization trick as $\mathbf{z} = \mu + \sigma \cdot \epsilon$, where $\epsilon \sim \mathcal{N}(0, I)$ and $\sigma = \exp\left(0.5 \cdot \log \sigma^2\right)$.

This sampled vector is concatenated with the condition vector and passed through the decoder, which mirrors the encoder in reverse with hidden layers of 32 and 64 units, and a final output layer matching the dimensionality of the input feature vector.

The model is trained using a composite loss function consisting of a reconstruction loss and a regularization term. The reconstruction loss is the MSE between the reconstructed and original feature vectors. Regularization is achieved via the Kullback–Leibler (KL) divergence between the learned latent distribution and the standard normal prior.

The total loss function is presented in Equation 5.

$$\mathcal{L}_{\text{cVAE}} = \text{MSE}(\hat{\mathbf{x}}, \mathbf{x}) + \beta \cdot \text{KL}(\mu, \log \sigma^2) \tag{5}$$

where $\beta = 1.0$ controls the regularization strength. The model was trained using the Adam optimizer with a learning rate of 0.001, batch size of 64, for 1500 epochs.

*b) Conditional Generative Adversarial Network (cGAN):* The cGAN architecture [31] comprises two main neural networks: a generator and a discriminator. The overall block diagram of cGAN is presented in Fig. 4. The generator is designed to create synthetic data samples that mimic the real tabular data. It takes as input a concatenated vector of a noise vector (sampled from a normal distribution) and a condition vector (derived from the one-hot encoded labels). This combined input is first passed through a linear layer that outputs 64 features, followed by a ReLU activation. The resultant activations are then transformed through subsequent layers that expand the dimensionality to 128 and finally map back to the original input dimensionality, thus generating a synthetic data sample.

Conversely, the discriminator functions as a binary classifier that aims to distinguish between real and fake samples. It too, receives a concatenated input of a data sample (either real or generated) and the condition vector. The input is processed through a series of linear layers with sizes 128 and 64, both activated by ReLU, and culminates in an output layer with a single unit activated by the Sigmoid function. This output represents the probability that the input sample is real.

The training procedure involves setting up both networks with the Adam optimizer (learning rate set to 0.001 for each). A binary cross-entropy loss function (BCE) is employed to quantify the differences between the discriminator's predictions and the ground truth labels (where real samples are labeled as 1 and fake samples as 0). During each training epoch, mini-batches of the data are used: the discriminator is first updated by computing the BCE loss for real samples (paired with their corresponding one-hot condition) and for the synthetic samples produced by the generator (also paired with the same condition). After updating the discriminator, the generator is updated by feeding its synthetic outputs through the discriminator and computing the BCE loss against the target of real labels. This adversarial training loop is repeated for 1500 epochs.

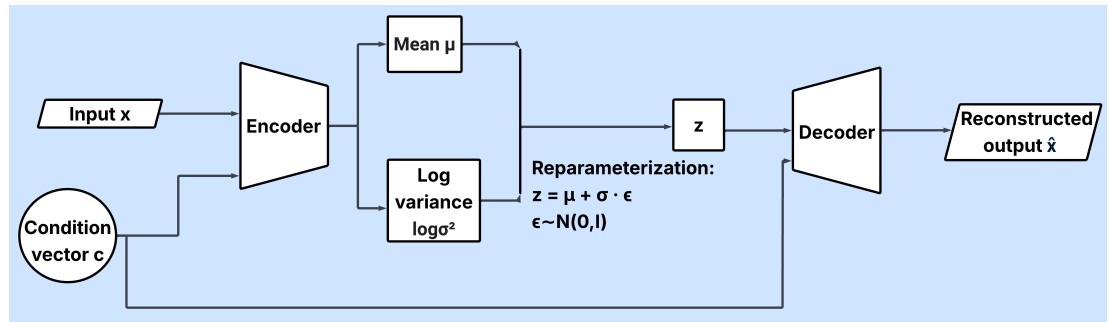

Fig. 3. Block diagram of cVAE. The model takes a feature vector $x$ and condition vector $c$ as input. The encoder outputs the mean $\mu$ and log-variance $\log \sigma^2$ of a latent Gaussian. A latent variable $z$ is sampled via the reparameterization trick. The decoder reconstructs $\hat{x}$ from $z$ and condition $c$.

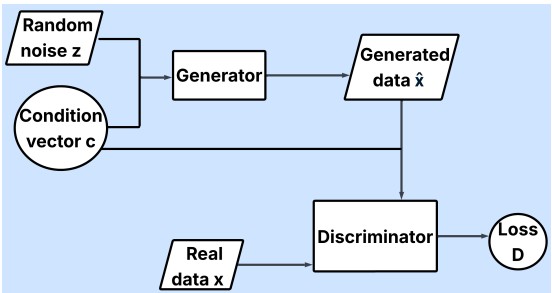

Fig. 4. Block diagram of cGAN. The generator receives a noise vector and condition vector to produce synthetic data. The discriminator is given real and generated samples, each paired with the same condition, and learns to distinguish between them.

## III. RESULTS

We begin our experiment to predict ISS using the cleaned, pre-processed dataset. Fig. 5 shows that the distribution of ISS is highly imbalanced, with most samples clustered at low values and a steep drop as severity increases, leaving few high-score cases.

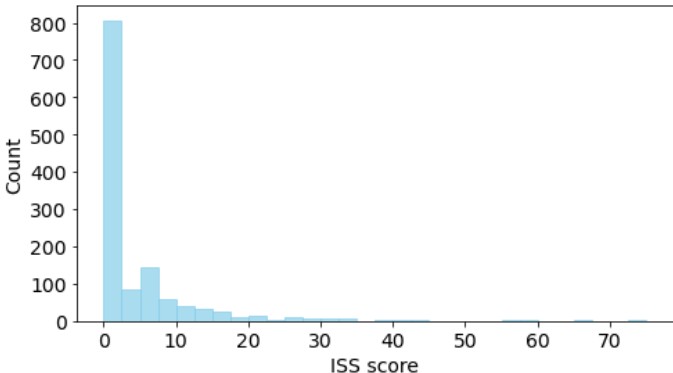

Fig. 5. Original ISS distribution. The data is highly right-skewed, with most cases clustered between 0 and 5 and a long tail extending up to 75.

We train and evaluate four ensemble regressors: RFR, GBR, XGBoost, and LightGBM. Performance results are shown in Table I. GBR performs the best, with the lowest MAE, MSE, and highest $R^2$, indicating strong predictive accuracy and variance explanation. XGBoost performs the worst, with the highest MAE, MSE, and lowest $R^2$. RFR and LightGBM have similar $R^2$, though RFR slightly outperforms in MAE and MSE. Overall, GBR is the most effective for ISS prediction, benefiting from residual learning.

However, since the task is based on a highly imbalanced dataset, the models tend to perform well in the dominant group (low ISS) but underperform in rare, high-severity cases due to limited data. Therefore, the performance of GBR with its comparatively high $R^2$ (0.78) is also not able to correctly capture the rare high-severity injuries. To overcome this challenge, we explore various data augmentation strategies designed to improve generalization across the full ISS range. These tailored strategies include resampling, interpolation-based, and noise-based methods. We also explore generative models such as cVAE and cGAN to synthesize realistic ISS data.

TABLE I
PERFORMANCE COMPARISON OF DIFFERENT REGRESSION MODELS

| Model | MAE | MSE | $R^2$ |
|---|---|---|---|
| Random Forest Regressor (RFR) | 2.40 | 17.54 | 0.76 |
| Gradient Boosting Regressor (GBR) | **2.24** | **16.03** | **0.78** |
| XGBoost Regressor | 2.70 | 20.35 | 0.72 |
| LightGBM Regressor | 2.42 | 17.79 | 0.76 |

The best-performing model from our experiments, GBR, is used as the baseline for evaluating different augmentation strategies. For each method, we train the GBR model on a corresponding augmented training set, allowing it to learn from the combination of the original and synthesized samples. To maintain fairness, the model performance is evaluated on the original held-out test set of size 255.

Table II summarizes the performance and size of the training dataset for each method. The baseline GBR model (1,017 samples) yields MAE = 2.24 and $R^2$ = 0.78. Transformation-based techniques (logarithmic, weighting) use the same data size but show degraded performance (MAE > 4.7, $R^2$ < 0.70). Resampling methods moderately increase dataset size; SMOGN expands it to 1,495 and yields improved performance (MAE = 1.98, $R^2$ = 0.89), while manual target-based upsampling (1,068 samples) performs even better (MAE = 1.67, $R^2$ = 0.90). The KDE variant performs worse (MAE = 2.51, $R^2$ = 0.74). Interpolation (mixup) and noise-based (Gaussian) augmentations add minimal data (1,017 and 1,034 samples), yielding moderate gains. Generative models produce

TABLE II
PERFORMANCE OF GBR MODEL TO PREDICT ISS USING DIFFERENT DATA AUGMENTATION STRATEGIES

| Category | Experiment | Training dataset size | MAE | MSE | $R^2$ |
|---|---|---|---|---|---|
| Baseline GBR | Without augmentation | 1017 | 2.24 | 16.03 | 0.78 |
| A. Transformation | Logarithmic transformation | 1017 | 4.7 | 24.32 | 0.70 |
|  | Weighting samples | 1017 | 5.3 | 27.20 | 0.65 |
| B. Resampling | SMOGN | 1495 | 1.98 | 8.37 | 0.89 |
|  | Upsampling-manual | 1068 | 1.67 | 7.30 | 0.90 |
|  | Upsampling-KDE | 1068 | 2.51 | 20.14 | 0.74 |
| C. Interpolation-based | Mixup Interpolation | 1017 | 2.45 | 18.71 | 0.77 |
| D. Noise-based | Gaussian Noise Injection | 1034 | 2.53 | 18.53 | 0.76 |
| E. Generative Models | Conditional VAE (cVAE) | 1517 | **1.53** | **7.19** | **0.94** |
|  | Conditional GAN (cGAN) | 1717 | 1.86 | 7.94 | 0.92 |

the largest datasets and best results: cVAE (1,517 samples) achieves top performance (MAE = 1.53, $R^2$ = 0.94), while cGAN (1,717) also performs well (MAE = 1.86, $R^2$ = 0.92). Compared to the recent study by [16], which used deep learning approaches (NMT and FFNN) and reported ISS accuracy of 79.8%, our model shows substantially higher performance.

Fig. 6 compares MAE, MSE, and $R^2$ across all augmentation methods. Transformation techniques perform the worst, resampling (manual and SMOGN) show improved accuracy, and generative approaches, particularly cVAE, lead in all metrics, validating their effectiveness for handling regression imbalance and improving generalization across the ISS spectrum. To visualize prediction quality, Fig. 7 compares actual vs. predicted ISS for baseline GBR and cVAE-augmented GBR. The baseline underpredicts high ISS values, while cVAE yields tighter alignment with the ideal line, especially for rare high-severity cases.

To demonstrate robustness across trauma severity levels, we performed residual error analysis (Fig. 8) across ISS bins: ISS≤15, 16≤ISS≤29, and ISS≥30. The high-severity bin (ISS≥30) showed the largest improvement, reducing residual standard deviation from 18.79 to 5.10 over the baseline GBR. Additionally, we implemented 5-fold cross-validation in conjunction with the same hold-out test. The final model from cross-validation is then evaluated on the hold-out test data, achieving $R^2$ of 0.934, which demonstrates comparable performance with the initial performance. Considering the importance of flagging high-risk cases, we performed binary classification of ISS>15 using the baseline GBR model and achieved 97.3% overall accuracy, which outperforms the accuracy of 94.0% reported in [16]. For the ISS>15 class, we obtained a precision of 0.945 and a recall of 0.929.

## IV. CONCLUSIONS

We presented an ML framework for predicting ISS using structured clinical and demographic data. Among four ensemble-based regression models, the GBR achieved the best baseline performance with an $R^2$ of 0.78. However, due to the highly imbalanced nature of the ISS distribution, baseline models struggled to generalize to rare, high-severity cases. To address this, we implemented a range of data aug-

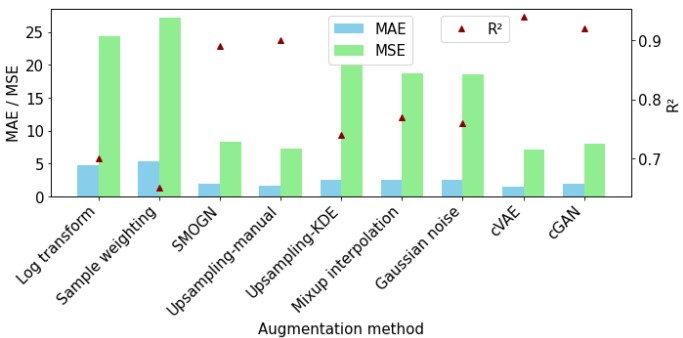

Fig. 6. Visual comparison of GBR performance across data augmentation strategies using MAE, MSE, and $R^2$. Bar plots show MAE (sky blue) and MSE (light green) on the left y-axis, while red triangles indicate $R^2$ values on the right y-axis. Augmentation methods are shown along the x-axis

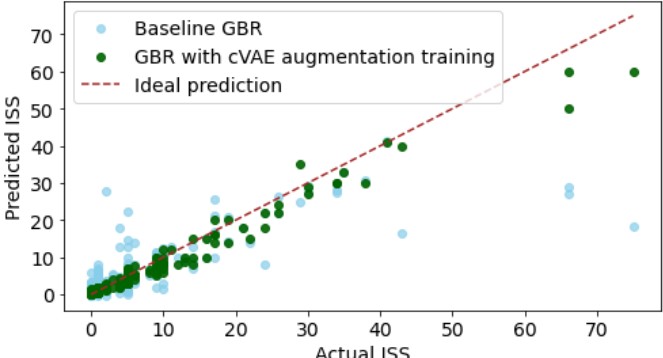

Fig. 7. Scatter plot comparing actual vs. predicted values for two GBR models in a ISS prediction regression task. The baseline GBR model (with no augmentation) is shown in light blue, while the GBR model trained with cVAE-based data augmentation is represented in dark green. Each point represents a prediction, and the red dashed line indicates the ideal $y = x$ case.

mentation techniques, including transformation, resampling, interpolation, and noise-based methods, as well as generative approaches using cVAE and cGAN. The cVAE-augmented GBR model achieves the highest predictive performance with an $R^2$ of 0.94 and MAE of 1.53, highlighting the effectiveness of generative data augmentation in improving regression accuracy under severe class imbalance. These findings underscore the potential of ML and advanced augmentation strategies to enhance trauma severity prediction and support data-driven

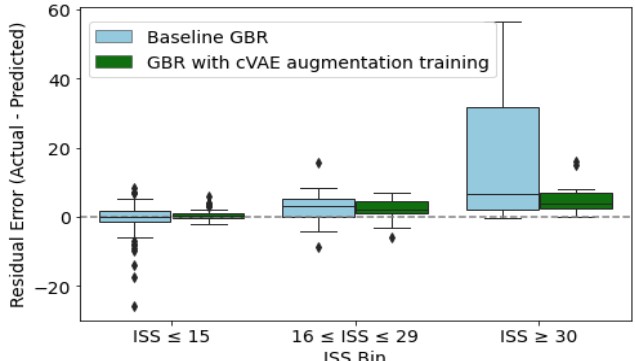

Fig. 8. Residual errors (Actual – Predicted) across ISS bins: ISS $\leq$ 15, $16 \leq$ ISS $\leq$ 29, and ISS $\geq$ 30.

clinical decision-making. Future work will focus on validating the model using triage-only features and external datasets to support real-time deployment, enhance generalizability across institutions, exploring other generative methods, and exploring downstream clinical utilization.

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
