# OpenReview forum: "Predicting Trauma Severity from Imbalanced Data Using Ensemble Regression and Generative Models"
_IEEE.org/EMBS/BHI/2025/Conference — BHI 2025_

### Official Review · Reviewer_x7QU · 2025-06-25
**Machine learning to predict injury severity prediction based on limited data**

**Confidence:** 5
**Clarity Of Writing:** excellent
**Clinical Significance:** great
**Methodological Novelty:** good
**Overall Rating:** 6

**Experiments And Results:**

good

**Questions For The Authors:**

1. In the introduction, the manuscript highlights the dependence of ISS calculation on imaging modalities such as CT, X-ray, and ultrasound. However, the dataset used in this study does not appear to incorporate imaging-derived features. Could the authors clarify this discrepancy and elaborate on how the model approximates ISS without these clinically essential inputs?
2. The introduction emphasizes the need to predict ISS even without the full diagnostic data. Was the validation dataset constructed using only features accessible at the point of initial triage, or was the model evaluated on the full dataset including retrospective information?
3. Did the authors conduct an analysis of false positives and false negatives, particularly in the high-ISS range? Understanding the types and clinical implications of prediction errors is critical for evaluating the model’s safety and reliability in triage settings.

**Strengths:**

1. The study thoroughly investigates and compares classical and generative augmentation techniques to address regression imbalance, including transformations, SMOGN, KDE, mixup, and noise injection.
2. Incorporating cVAE and cGAN for ISS data synthesis is a novel approach in trauma prediction and demonstrably improves model generalization to rare high-severity cases.
3.The paper offers a clear and reproducible workflow, detailing data preprocessing, feature engineering, model training, and evaluation metrics.

**Summary Of The Paper:**

This study proposes a machine learning framework to predict Injury Severity Score (ISS) using structured clinical, demographic, and vehicle data from a trauma center. The authors compare four ensemble regressors and introduce various data augmentation techniques to address severe data imbalance, particularly in high-severity trauma cases. The best model, a Gradient Boosting Regressor trained on cVAE-augmented data, achieved an R² of 0.94, highlighting the potential of generative augmentation to enhance performance in skewed clinical datasets.

**Weaknesses:**

1. The study uses a relatively small, single-center dataset (N=1273), which may limit the generalizability of results across institutions or populations. The dataset from other institutions may be attributed differently, which can be challenging for generalizability.
2. The work focuses exclusively on predicting ISS, without exploring downstream clinical utility, which could strengthen clinical relevance.
3. No validation on an external dataset was performed, raising concerns about potential overfitting, especially given the synthetic data augmentation.

---

### Official Review · Reviewer_o6n2 · 2025-07-08
**ISS regressor with interesting comparisons of upsampling methods to address label imbalance**

**Confidence:** 5
**Clarity Of Writing:** excellent
**Clinical Significance:** good
**Methodological Novelty:** fair
**Overall Rating:** 6
**Final Rating:** 7

**Experiments And Results:**

excellent

**Questions For The Authors:**

None

**Strengths:**

1. The pipeline in Figure 1 is clear and well-structured; adding more distinct colors would enhance its readability.

2. The machine-learning workflow is thoughtfully designed and well-suited to predicting trauma severity from heterogeneous data.

3. The paper provides a thorough evaluation of upsampling methods, offering a strong comparative analysis of traditional and generative approaches.

**Summary Of The Paper:**

This paper presents an end-to-end framework for predicting the Injury Severity Score (ISS) from routinely collected clinical, demographic, and vehicle data using ensemble regression and generative augmentation. After evaluating four tree-based regressors (Random Forest, Gradient Boosting, XGBoost, LightGBM) on a trauma‐center data, the authors address the severe skew toward low-severity cases with a variety of augmentation strategies, including two conditional generative models. Augmenting with a cVAE boosts performance to R² = 0.94 and MAE = 1.53, demonstrating that synthetic high-severity samples substantially improve regression accuracy under class imbalance.

**Weaknesses:**

1. Figure 6 adds limited value; consider removing it and giving Table II more prominence in the main text.

2. To contextualize your results, include a comparison with other ISS-prediction or trauma-severity methodologies from the literature.

3. Figure 2 is barely readable.

4. I would recommend running 5-fold cross validation and have a hold-out test set for testing upsampling strategies.

---

### Official Review · Reviewer_pfn7 · 2025-07-17
**Predicting Trauma Severity from Imbalanced Data Using Ensemble Regression and Generative Models**

**Confidence:** 5
**Clarity Of Writing:** good
**Clinical Significance:** great
**Methodological Novelty:** good
**Overall Rating:** 5

**Experiments And Results:**

good

**Questions For The Authors:**

How do the models generalize to data from different institutions or regions?
External validation on a different trauma dataset would significantly boost confidence in the framework’s real-world applicability.

What measures were taken to avoid overfitting in cVAE and cGAN given the small dataset?


Why is there no comparison to traditional trauma severity scoring methods like TRISS?

What was the rationale behind choosing cVAE over other structured tabular generative models?
While cVAE and cGAN are standard, methods like CTGAN or TabDDPM might offer better performance on tabular data. A brief justification would be helpful.

**Strengths:**

1- Tackles a relevant and clinically important problem (ISS prediction) with clear motivation.

2- Uses a diverse set of classical and generative augmentation strategies to address data imbalance.

3- Achieves significant performance improvement with cVAE (R² = 0.94), demonstrating clear value added.

4- Provides detailed implementation, model architecture, and evaluation metrics with good visualizations.

**Summary Of The Paper:**

This paper presents a machine learning framework to predict Injury Severity Score (ISS) — a critical trauma severity metric using structured clinical, demographic, and vehicle data. The authors evaluate four ensemble regressors and find Gradient Boosting Regressor (GBR) performs best (R² = 0.78). Given the right-skewed ISS distribution, they apply various data augmentation strategies (log transforms, SMOGN, interpolation, Gaussian noise, etc.) and develop generative models (cVAE and cGAN) to enhance model robustness, especially for high-severity cases. The cVAE-augmented GBR achieves the highest performance (R² = 0.94), showing the value of synthetic data in handling imbalanced regression tasks.

**Weaknesses:**

Major Comments:

1- The dataset is relatively small (1,273 samples), raising concerns about overfitting, especially with generative models.

2- No external validation or generalizability assessment is presented (e.g., different trauma centers, hospitals).

3- There is no clinical interpretation or error analysis (e.g., how often are high-risk patients underpredicted?).

4- Hyperparameter tuning details for ensemble models and augmentation methods are missing.

5- Impact of augmentation on interpretability or clinical reliability is not discussed.

Minor Comments:

1- Fig. 5–8 could benefit from clearer axis labeling and consistent color schemes.

2- Table II is valuable but could be expanded with standard deviations or confidence intervals for robustness.

3- Figures are embedded as low-resolution images, and axis labels are missing or hard to read. This hinders interpretability.

---

### Official Review · Reviewer_LRN8 · 2025-07-17
**Predicting Trauma Severity from Imbalanced Data Using Ensemble Regression and Generative Models**

**Confidence:** 2
**Clarity Of Writing:** good
**Clinical Significance:** good
**Methodological Novelty:** poor
**Overall Rating:** 5

**Experiments And Results:**

good

**Questions For The Authors:**

Authors mentioned:
Despite these advancements, an important gap remains that no ML-based model has been specifically developed to predict the ISS or trauma severity. Though literature has emphasized the importance of predicting ISS [14], direct ISS prediction has not yet been systematically addressed.

I found the following published literatures:
1. Predicting ISS from ICD Codes: https://pubmed.ncbi.nlm.nih.gov/39485495/
A study compared deep-learning models to translate ICD diagnosis codes into ISS scores using neural machine translation (NMT) and feed-forward neural networks (FFNN).
The top-performing model—NMT with an indirect approach (via AIS codes)—achieved 94% accuracy in predicting whether ISS ≥ 16, and correctly matched exact ISS scores about 79.8% of the time.

2. Mortality Prediction Models Using ISS & Clinical Data: https://pubmed.ncbi.nlm.nih.gov/34876644/
https://pubmed.ncbi.nlm.nih.gov/34419627/
A deep-learning model developed in the Korean Trauma Data Bank outperformed traditional ISS and NISS systems, achieving higher accuracy (85.05%) and AUROC (0.9084) using region-based AIS inputs and a DNN architecture


Another study developed the Spinal Cord Injury Risk Score (SCIRS), which predicted mortality with AUC ~0.86, significantly outperforming

3. Real-Time EHR-Based Mobility and Mortality Prediction: https://pubmed.ncbi.nlm.nih.gov/34932043/
A trauma registry evaluated the Epic Deterioration Index (EDI) model, finding it achieved AUC of 0.98 for mortality prediction—surpassing ISS and NISS performance

**Strengths:**

he key contributions of this work include: (1) the development of a machine learning pipeline to predict Injury Severity Score (ISS) using a motor vehicle collision dataset, where ensemble-based regressors are compared and Gradient Boosting Regressor (GBR) achieves the best baseline performance (R² = 0.78); (2) an in-depth analysis of dataset imbalance and its effect on regression accuracy, followed by the application of diverse data augmentation strategies such as log transformations, SMOGN, KDE-based upsampling, and mixup; and (3) the design and implementation of two deep generative models, cVAE and cGAN, to synthesize underrepresented high-severity cases, with cVAE-enhanced training significantly improving performance (R² = 0.94). These contributions demonstrate the potential of generative augmentation to address skewed regression targets in clinical data.

**Summary Of The Paper:**

This study proposes a machine learning framework to predict the Injury Severity Score (ISS) using structured clinical, demographic, and vehicle data. The authors evaluate four ensemble-based regression models and identify Gradient Boosting Regressor (GBR) as the best performer. To address significant class imbalance in the dataset, they employ various augmentation strategies, including advanced generative models like cVAE and cGAN. Notably, cVAE-augmented training leads to the highest predictive accuracy, highlighting the effectiveness of generative augmentation in improving regression performance for skewed clinical data.

**Weaknesses:**

Authors mentioned:
Despite these advancements, an important gap remains that no ML-based model has been specifically developed to predict the ISS or trauma severity. Though literature has emphasized the importance of predicting ISS [14], direct ISS prediction has not yet been systematically addressed.

I found the following published literatures:
1. Predicting ISS from ICD Codes: https://pubmed.ncbi.nlm.nih.gov/39485495/
A study compared deep-learning models to translate ICD diagnosis codes into ISS scores using neural machine translation (NMT) and feed-forward neural networks (FFNN).
The top-performing model—NMT with an indirect approach (via AIS codes)—achieved 94% accuracy in predicting whether ISS ≥ 16, and correctly matched exact ISS scores about 79.8% of the time.

2. Mortality Prediction Models Using ISS & Clinical Data: https://pubmed.ncbi.nlm.nih.gov/34876644/
https://pubmed.ncbi.nlm.nih.gov/34419627/
A deep-learning model developed in the Korean Trauma Data Bank outperformed traditional ISS and NISS systems, achieving higher accuracy (85.05%) and AUROC (0.9084) using region-based AIS inputs and a DNN architecture


Another study developed the Spinal Cord Injury Risk Score (SCIRS), which predicted mortality with AUC ~0.86, significantly outperforming

3. Real-Time EHR-Based Mobility and Mortality Prediction: https://pubmed.ncbi.nlm.nih.gov/34932043/
A trauma registry evaluated the Epic Deterioration Index (EDI) model, finding it achieved AUC of 0.98 for mortality prediction—surpassing ISS and NISS performance

---

### Official Review · Reviewer_uXPH · 2025-07-22
**Nice idea to guess trauma score, but need more check**

**Confidence:** 3
**Clarity Of Writing:** great
**Clinical Significance:** good
**Methodological Novelty:** good
**Overall Rating:** 5

**Experiments And Results:**

good

**Questions For The Authors:**

Can you test model on another center or later year to show generalization?
How many high‑severity (ISS > 30) cases in original vs synthetic?
Did you compare to simple linear or clinical rule‑based baseline?

**Strengths:**

ISS helps triage quick.
Clear pipeline: preprocess, features, many regressors, then augmentation list.
Try many imbalance methods and compare; table with MAE/MSE/R² easy to read.
Generative trick works big (0.78 → 0.94). This is interesting for small rare data.
Feature importance explains which factors matter (LOS, intubation).

**Summary Of The Paper:**

Authors use one trauma file with 1273 rows and 62 columns from one Level‑1 center. They want to predict Injury Severity Score only from normal registry numbers, not imaging.
First they test four tree ensembles. Gradient Boosting Regressor is best, R² = 0.78 on hold‑out test.
Because ISS very unbalanced to low values, they try many data‑boost tricks: SMOGN, mix‑up, up‑sample, etc. They also build two generator models (cVAE, cGAN). With cVAE data the GBR jumps to R² = 0.94. Paper says generative boost is top.

**Weaknesses:**

Only one hospital dataset, small (1273). May over‑fit to local practice.
No external or temporal test, so 0.94 may drop outside.
ISS > 15 threshold not discussed; regression nice, but clinic needs high‑risk flag.
Calibration / error for high ISS not shown; R² hides that.